# Serological, Virological Investigation and Hepatic Injury Evaluation for Hepatitis E Virus in Hunting Dogs

**DOI:** 10.3390/pathogens11101123

**Published:** 2022-09-29

**Authors:** Angelica Bernardini, Maria Irene Pacini, Niccolò Fonti, Mario Forzan, Veronica Marchetti, Maurizio Mazzei

**Affiliations:** Department of Veterinary Sciences, University of Pisa, Viale delle Piagge 2, 56124 Pisa, Italy

**Keywords:** HEV, hunting dogs, serology, fecal swab, infectious disease

## Abstract

Hepatitis E virus (HEV) is a quasi-enveloped single-stranded positive-sense RNA virus belonging to the Orthohepevirus A genus within the *Hepeviridae* family. The most common transmission route of this virus is fecal–oral, although zoonotic transmission by contact with infected animals has also been described. In this study, 80 sera and rectal swabs were collected from dogs during the 2019/2020 and 2020/2021 wild boar hunting season in Tuscany. All dogs were submitted for serological screening to detect the presence of anti-HEV antibodies. To evaluate the circulation of HEV, rectal swabs from both seropositive dogs and dogs living in the same kennels were examined by One-Step RT-qPCR. In addition, the presence of markers of hepatic damage in dogs’ sera was investigated. Results indicated the presence of anti-HEV antibodies in 4/80 subjects (5%). However, neither HEV RNA nor signs of hepatic damage were found. In conclusion, although HEV can stimulate a specific immuno-response in dogs, this species does not seem to play an important role in HEV epidemiology.

## 1. Introduction

Hepatitis E is an emerging infection and it is considered an important disease worldwide [1]. Eight genotypes are included in this genus: genotypes 1 and 2 infect only humans and are mainly diffused in developing countries; genotypes 3 and 4, endemic in both developing and industrialized countries, are zoonotic and have been detected in humans and various animal species, in particular in pigs and wild boar, which are considered the main reservoirs; genotypes 5 and 6 have been identified exclusively in wild boar; genotypes 7 and 8 have been recently reported in dromedaries and camels, respectively [2]. HEV-3, the most common genotype in Europe, has been divided into 11 subtypes (3a–3j and 3ra). However, despite this classification, many subtypes remain unassigned due to the presence of new emerging strains [3,4,5,6]. 

In humans, HEV can cause acute self-limiting hepatitis, which in immunocompromised individuals could lead to chronic hepatitis. It has also been documented that mortality rates reach 25% in pregnant women with chronic hepatitis [1,7]. 

The most common transmission route is fecal–oral and through the consumption of contaminated water and food, although the zoonotic transmission by direct and indirect contact with infected animals, in particular swine, has also been described [8,9]. Various studies have also reported a high prevalence of HEV in veterinarians, farmers, slaughterhouse staff, and hunters, supporting an increased infection risk for these professional activities [10,11,12]. 

By serological testing, anti-HEV antibodies have been detected in several domestic and wild animals [13,14]. The presence of HEV in the swine industry and in wild boar in central Italy and Tuscany has been investigated, reporting a high level of seroprevalence and virological evidence over many years. In detail, the HEV herd seroprevalence in pig farms in northern Italy has been reported as 75.6% [15]. Other studies conducted in 2015 by Caruso and colleagues on wild boar, reported a prevalence of 3.7% in northwest Italy [16], while Martelli and colleagues reported a prevalence of 25% in northeast Italy [17]. Furthermore, between 2016 and 2017, HEV-3c and HEV-3f RNA has been identified in Northern Italy in 52.2% of wild boar livers [18]. Virological surveys to detect HEV in 611 liver samples of wild boar shot during three consecutive hunting seasons (2016–2020 in the Umbria-Marches Apennines found a prevalence of 2.45%, while no positivity was recorded for serum samples [19]. A study conducted between 2011 and 2014 in central Italy identified 16.3% of wild boar liver samples as HEV-positive. [20]. The HEV seroprevalence in wild boar from previous studies conducted in Tuscany has been reported as 56.2%, with a molecular positivity of fecal samples of 9.4% [21]. Moreover, several studies conducted on wild animals roaming in Tuscany have also shown a constant presence of HEV [21,22,23,24,25,26]. Regarding the seroprevalence of HEV in dogs in Europe, it has been described as ranging from 56.6% to 14.3% [21,27,28,29]. However, despite this, HEV RNA has not yet been identified. Considering that wild boars are an important reservoir for HEV and that their hunting is a highly practiced activity in Tuscany, this study aimed to assess the presence of HEV antibodies and RNA in serum and rectal swabs of hunting dogs. Furthermore, since no data about the clinical implications of HEV infection in dogs are available, the evaluation of liver function parameters in the serum of the HEV-seropositive animals has also been performed.

## 2. Results

A total of 80 hunting dogs were sampled, 47/80 male (59%) and 33/80 female (41%); 10/80 were less than 2 years old, 43/80 were between 2 and 5 years old and 27/80 were over 5 years old. Eighteen dogs were from Lucca (7 from 2 kennels, 11 alone), 30 from Pisa (in 6 kennels), and 32 from the province of Pistoia (23 in 3 kennels and 9 alone). The serological analysis indicated that 4 out of 80 sera (5%) scored positive for anti-HEV antibodies. With regard to the provenience of the seropositive animals, 3 came from Pistoia (2 of which were from the same kennel), and 1 from Pisa. Regarding age and sex, 2 were male and 2 were female, 3 were aged between 2 and 5 years, and 1 was over 5 years old. RNA was extracted from rectal swabs of seropositive dogs and those living in the same kennel (17/80). The spectrophotometric analyses conducted on the extracted RNA showed that all the samples were suitable for subsequent molecular investigations (mean concentration 67.34 ± 32.94 ng/µL). The PCR targeting the reference gene 18s gave a positive result in all tested samples. The RT-qPCR analyses gave a negative result for all the samples tested. The biochemistry analyses, performed on the same panel of 17 animals subjected to molecular analysis, showed values in line with the reference ranges for all the evaluated parameters (ALP 99.7 + 42.9 U/L (RR 45–50 U/L); GGT 2.26 + 1.7 (RR 2–11 U/L); ALT 30.9 + 12.8 U/L (RR 20–70 U/L); AST 20.6 + 12.8 (RR 15–40 U/L). 

## 3. Discussion

There is very limited scientific information about HEV infection in dogs and the role they play in the epidemiology of the virus. To date, HEV RNA has not been detected in any of the previous studies performed worldwide on dogs. In addition, there are only few and variable data regarding the HEV seroprevalence in dogs, ranging between 0% and 60.1% [27]. This high variability could be due to the spread of the virus in the study areas and on the variety of living conditions and attitudes of the animals included in the different studies. Most of the studies were conducted on pets, while others were on stray dogs or sheep dogs, and only one on hunting dogs [27]. It is noteworthy that none of the previous studies conducted a molecular investigation on hunting dogs, despite the fact that hunting can be considered a high-risk activity due to the possibility of contact with wild boar. 

In the present study, we investigated the presence of HEV antibodies and HEV-RNA in dogs used for hunting in an area where HEV circulation was already identified by serological and molecular assays in wild boars, rabbits, and wild ruminants [21,23,24]. 

For initial screening purposes, all animals were serologically tested using a commercial ELISA kit able to detect all anti-HEV antibodies (IgM, IgG, and IgA), and therefore both recent and past infections could have been identified. Our results indicate a 5% seropositivity which is lower than what has been reported in previous European studies [21,27,28,29] Ingestion of contaminated food is one of the main transmission routes of HEV in suids and humans [30,31], and probably also in dogs [32,33,34]. Indeed, feeding dogs with kitchen waste or infected animal offal has been identified as an HEV risk factor in HEV transmission practice [34,35]. 

Considering that the studied population was composed exclusively of dogs used for wild boar hunting, and therefore theoretically at greater risk of contracting the infection, the low seroprevalence that we obtained was unexpected, especially when compared to previous studies. A possible explanation for this result is that Tuscan hunters do not usually feed their dogs with pieces of carcass because of the high prevalence of pseudorabies in wild boar [25,36].

Molecular analysis performed on seropositive animals and those cohabiting with them did not identify any positive samples among the 17 rectal swabs tested. This result agrees with previous studies, in which HEV RNA has never been found in dog serum [32,37,38], body cavity transudate [28], feces [37], or liver [13], even after experimental infection [32]. In a recent study, HEV RNA has been identified in a wolf (*Canis lupus italicus*) rectal swab, which could be due to the predation of an infected animal [39]. Following this evidence, the negative results from serum samples, and the result on sera samples even after experimental infection, we considered rectal swabs a suitable sample for our study.

Although the absence of HEV-RNA in domestic dogs’ feces indicates short or limited HEV viremia, we cannot exclude that canines could still contribute to HEV spread in the environment with a role in the virus epidemiological cycle, especially considering the close relationship between dogs and humans.

Although worldwide reported seropositivity denotes the susceptibility of dogs to HEV, there are no data or any clinical evidence of infection in positive animals. In our study, no anatomical alterations were found for any of the tested parameters indicating the absence of liver damage in infected subjects that had no clinical signs of infection. However, this result is in line with what has also been observed in all the other HEV susceptible species that seem to be infected mainly asymptomatically, except for primates and avian species [14,31,40]. 

These data could indicate that in domestic dogs, HEV stimulates a specific immune response, but the main HEV infection outcome is either abortive or subclinical, therefore not posing a health risk in this species, nor a significant public health sanitary risk. 

## 4. Materials and Methods

### 4.1. Sampling

During the 2019/2020 and 2020/2021 wild boar hunting seasons (Regolamento di attuazione della legge regionale 12 gennaio 1994 n◦ 3 DPGR 48/R/2017), healthy dogs used for hunting activity in the provinces of Lucca, Pisa, and Pistoia (Tuscany, Italy) were included in the study.

Serum and a rectal swab were collected from each dog at veterinary clinics during clinical visits and transported to the Department of Veterinary Science (University of Pisa, Italy) for serological and molecular analysis. Age, sex, living area, and clinical history of each sampled animal were recorded. Swabs were frozen at −20 °C until further studies were performed.

The Ethical Approval was provided by the Institutional animal care and use committee of the University of Pisa (prot N. 23507/2015).

### 4.2. Serological Analysis

All animals included in the study were submitted for serological screening. Individual serum samples were analyzed using a commercial enzyme-linked immunoassay (HEV Ab version ULTRA, DIA.PRO, Milano, Italy), following the manufacturer’s instructions. This double-antigen sandwich ELISA assay is based on a recombinant ORF-2 encoded protein, which is highly conserved in all HEV genotypes [41], allowing qualitative detection of total anti-HEV antibodies(IgG, IgM, and IgA) in both human and non-human serum and plasma samples. Optical density (OD) was quantified by spectrophotometric reading at 450 nm (Multiskan FC; Thermo Fisher Scientific, Waltham, MA, USA).

For each sample, the ratio between the OD of the sample (Sn) and the cut-off (Ratio Sn/CO) was expressed and interpreted as instructed by the manufacturers.

### 4.3. Molecular Analysis

ELISA-positive dogs and those that have potentially come into close contact with them (living in the same kennel) were included in the molecular analysis. Rectal swabs were thawed and soaked in 400 µL of sterile Phosphate Saline Buffer (PBS) in 2 mL eppendorfs, incubated for 2 h at room temperature (RT), and finally centrifuged at 3000× *g* for 10 min. Total RNA was extracted from the supernatant using the QIAamp Viral RNA Mini Kit (Qiagen, Hilden, Germany) following the manufacturer’s instructions. All of the RNA was quantified using NanoDrop (Thermo Fisher Scientific, Waltham, MA, USA) and a series of RNA was tested by PCR amplification of a reference gene [42]. A broad-spectrum RT-qPCR was performed on 2 µL of purified RNA following a previously described Taq-Man protocol targeting a 69-bps fragment of ORF3 [43]. RT-qPCR was carried out using the Luna Universal Probe One-Step RT-qPCR Kit (NEB, Ipswich, MA, USA). To achieve absolute quantification of the HEV RNA, a calibration standard curve was generated, as described elsewhere [24].

Each sample was assayed in duplicate and wild boar HEV RNA-positive samples, detected from the same area in a previous study [24], were used as positive controls.

### 4.4. Clinical Pathology

To evaluate possible liver damage due to HEV infection, 200 µL of sera from seropositive dogs were used to quantify a set of markers of hepatic injury: aspartate aminotransferase (AST), alanine transaminase (ALT), gamma glutamyl-transferase (GGT), and alkaline phosphatase (ALP) activities. 

Analyses were performed at the Laboratory of Veterinary Clinical Pathology of the Veterinary Teaching Hospital (Department of Veterinary Science, University of Pisa, Italy) using a spectrophotometric and immunoturbidimetric analyzer (Liasys Analyzer Medical System-AMS, Rome, Italy). 

## 5. Conclusions

Our serological results provide evidence of HEV infection in hunting dogs, but the absence of RNA in feces and a lack of clinical evidence suggest a very limited or absent viremia. On the other hand, we are not in the position to completely rule out domestic dogs playing a relevant epidemiological role in HEV transmission. However, considering the close relationship between dogs and humans and the limited number of samples investigated, further studies considering breed variety or HEV subtype is a key point to fully understand if these animals are susceptible and, in a limited way, possible virus shedders.

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
