# Peer review of "Serological, Virological Investigation and Hepatic Injury Evaluation for Hepatitis E Virus in Hunting Dogs"

_pathogens, 2022, doi:10.3390/pathogens11101123_

Round 1

Reviewer 1 Report

Since the studied population was composed of dogs used for hunting wild boars, considered one of HEV's main wild reservoirs, the authors omitted essential data: the incidence/prevalence of HEV RNA/antibodies in wild boars from Central Italy / Tuscany. Recent studies have reported high positivity rates of HEV RNA in specific Italian regions, including Central Italy (Di Pasquale et al., 2019). Moreover, since food-borne cases of HEV infection have been detected in Central Italy, data regarding HEV detection in wild boars must be investigated. In the present study, the authors assumed that the studied population is from an HEV endemic area. Besides, the authors assumed hunting (particularly wild boar) is a high-risk activity. Why did the authors not include a sampling of wild boars from the same region (Luca, Pisa, and Pistoia)? One would strongly advise authors to conduct such an investigation to compare the results found in hunting dogs.

Regarding the methodology, is the rectal swab adequate sampling? The authors did not present any reference showing that HEV RNA has already been detected in frozen rectal swabs. By the way, rectal swabs were frozen until HEV RNA detection by RT-qPCR. Have samples been submitted to previous freezing-thawing procedures? It is important to emphasize that the RNA quality was not affected by any other factor that could impair the RNA yield.

I agree with the authors that domestic dogs may play a role in HEV shedding; however, I am afraid I have to disagree that wild boars' hunting dogs do not represent a subject with an epidemiological role for transmission in an HEV endemic region. At least, authors should not assume that, considering the small number of samples investigated and the method's limitations.

Author Response

Reviewer 1

Since the studied population was composed of dogs used for hunting wild boars, considered one of HEV's main wild reservoirs, the authors omitted essential data: the incidence/prevalence of HEV RNA/antibodies in wild boars from Central Italy / Tuscany. Recent studies have reported high positivity rates of HEV RNA in specific Italian regions, including Central Italy (Di Pasquale et al., 2019). Moreover, since food-borne cases of HEV infection have been detected in Central Italy, data regarding HEV detection in wild boars must be investigated. In the present study, the authors assumed that the studied population is from an HEV endemic area. Besides, the authors assumed hunting (particularly wild boar) is a high-risk activity. Why did the authors not include a sampling of wild boars from the same region (Luca, Pisa, and Pistoia)? One would strongly advise authors to conduct such an investigation to compare the results found in hunting dogs.

Authors: Following the reviewer’s indication we implemented introduction section with a more detailed description of HEV wild-boar epidemiology in Italy. (line 55-75)

Regarding the methodology, is the rectal swab adequate sampling? The authors did not present any reference showing that HEV RNA has already been detected in frozen rectal swabs. By the way, rectal swabs were frozen until HEV RNA detection by RT-qPCR. Have samples been submitted to previous freezing-thawing procedures? It is important to emphasize that the RNA quality was not affected by any other factor that could impair the RNA yield.

Authors: All rectal swab RNA successfully passed quantity/quality control by nanodrop (mean concentration value of 67.34 +/- 32.94 ng/ul). Moreover a PCR amplifying a reference gene was also performed with positive result on a series of RNA (7 samples). All samples tested resulted positive. Material and methods (line 192) and results (line 100-102) were implemented.

A image of PCR performed on a series of RNA  sample amplifying reference gene is attached for the reviewer only. (see author notes file)

I agree with the authors that domestic dogs may play a role in HEV shedding; however, I am afraid I have to disagree that wild boars' hunting dogs do not represent a subject with an epidemiological role for transmission in an HEV endemic region. At least, authors should not assume that, considering the small number of samples investigated and the method's limitations.

Authors: Since no evidence of HEV shedding has been demonstrated in dogs in our study but also in other previous studies, even after experimental infection (Liu et al 2019 reference number 32), it is possible to hypothesize that dogs do not play a major role in the epidemiological cycle of HEV even in consideration that HEV shedding was documented to occur in many other animal species. According to reviewer’ suggestion we revised the sentence (line 213)

Reviewer 2 Report

The authors assessed the presence of HEV antibodies in serum of hunting dogs and tried to detect HEV RNA in 58 fecal swabs of those by RT-qPCR. The serological analysis indicated that 4 out of 80 sera (5%) scored positive for anti-HEV antibodies. The RT-qPCR analyses resulted negative for all the samples tested. The authors concluded that serological results provide evidence of HEV exposure of hunting dogs. This study provides interesting results and speculations.

Comments

In line 22 in abstract

The authors wrote “probably due to a short or very limited viremia.”

The authors do not have any evidence of the description. I recommend to delete the part.

The detection of HEV RNA from the feces does not always denote HEV infection, which means virus replication in dogs. The presence of HEV in the faecal content may represent the consequence of an ingestion of faeces containing the virus (See the following three studies). The authors collected serum. Why the authors did not try to detect HEV RNA in serum directly? After all, the authors could indicate no ongoing HEV infection. So please add some explanation. This is not a fatal problem.

De Sabato L, Ianiro G, Monini M, et al. Detection of hepatitis E virus RNA in rats caught in pig farms from Northern Italy. Zoonoses Public Health. 2019; doi:10.1111/zph.12655.

Grierson S, Rabie A, Lambert M, et al. HEV infection not evident in rodents on English pig farms. Veterinary Record. 2018;182:81-.

Kanai Y, Miyasaka S, Uyama S, et al. Hepatitis E virus in Norway rats (Rattus norvegicus) captured around a pig farm. BMC Res Notes. 2012;5:4.

Author Response

Reviewer 2

The authors assessed the presence of HEV antibodies in serum of hunting dogs and tried to detect HEV RNA in 58 fecal swabs of those by RT-qPCR. The serological analysis indicated that 4 out of 80 sera (5%) scored positive for anti-HEV antibodies. The RT-qPCR analyses resulted negative for all the samples tested. The authors concluded that serological results provide evidence of HEV exposure of hunting dogs. This study provides interesting results and speculations.

Comments

In line 22 in abstract

The authors wrote “probably due to a short or very limited viremia.”

The authors do not have any evidence of the description. I recommend to delete the part.

Authors: Following indication the part has been deleted (line 23)

The detection of HEV RNA from the feces does not always denote HEV infection, which means virus replication in dogs. The presence of HEV in the faecal content may represent the consequence of an ingestion of faeces containing the virus (See the following three studies). The authors collected serum. Why the authors did not try to detect HEV RNA in serum directly? After all, the authors could indicate no ongoing HEV infection. So please add some explanation. This is not a fatal problem.

Authors: Fecal swab were used instead of serum in view of a previous study in which HEV RNA was not detected in serum even after experimental infection of dogs despite an established immune response. Moreover, the only HEV RNA detection in the canis genus has been found in fecal sample (line 146-148) Therefore, when designing the experimental procedure we decided to use the described methodological approach.

Round 2

Reviewer 1 Report

After reviewing the revised manuscript, some comments must be made: 

1) Methods and Results are presented in more detail, as advised. However, an additional investigation aiming to assess HEV seroprevalence would substantially improve the Study Design.

2) Since the authors have added content to the sections Introduction, Methods, and Results, an extensive English language review is required. 

3)  Since the authors did not consider conducting additional investigation of HEV prevalence among wild boars from the studied region (it would substantially improve the Study Design), the article type should be changed to a Short Communication.

Author Response

Dear reviewer,

following the  indication, and due to the impossibility of conduct additional investigation of HEV prevalence among wild boars, we decided to change the article type to short communication. 
English language has been revised. 

Reviewer 2 Report

Now, the authors appropriately revised their manuscript.

Author Response

dear reviewer

thank for your comment on MS

sincerely

Maurizio Mazzei

Round 3

Reviewer 1 Report

In the previous report, the authors were advised to shorten the manuscript. However, extensive revision and editing are still needed. The authors must send the manuscript to an English native speaker for better comprehension of the whole  manuscript.

Author Response

Dear reviewer,

thank you for this further round of revision

Manuscript ID: pathogens-1874263
Type of manuscript: Short communication
Title: Serological, virological investigation and hepatic injury evaluation for Hepatitis e virus in hunting dogs
Authors: Angelica Bernardini, Maria Irene Pacini, Niccolò Fonti, Mario Forzan, Veronica Marchetti, Maurizio Mazzei *

Following indications we sent the MS to an English native speaker and corrections were made throughout the manuscript.

Moreover, we shortened the manuscript by removing parts that were not strictly essential even if there are not very strict formatting requirements for short communications.

Sincerely,

Maurizio Mazzei